# MMTSA: Multimodal Temporal Segment Attention Network for Efficient Human Activity Recognition

## Abstract

Multimodal sensors (e.g., visual, non-visual, and wearable) provide complementary information to develop robust perception systems for recognizing activities. However, most existing algorithms use dense sampling and heterogeneous subnetwork to extract unimodal features and fuse them at the end of their framework, which causes data redundancy, lack of multimodal complementary information and high computational cost. In this paper, we propose a novel multi-modal neural architecture based on RGB and IMU wearable sensors (e.g., accelerometer, gyroscope) for human activity recognition called Multimodal Temporal Segment Attention Network (MMTSA). MMTSA first employs a multimodal data isomorphism mechanism based on Gramian Angular Field (GAF) and then applies a novel multimodal sparse sampling method to reduce redundancy. Moreover, we propose an inter-segment attention module in MMTSA to fuse multimodal features effectively and efficiently. We demonstrate the importance of imu data imaging and attention mechanism in human activity recognition by rigorous evaluation on three public datasets, and achieved superior improvements (11.13% on the MMAct dataset) than the previous state-of-the-art methods.

## 1 Introduction

With the increased interests of wearable technologies and the development of deep learning, human activity recognition (HAR) has recently attracted widespread attention in human-computer interaction, healthcare, and multimedia analysis. Uni-modal (e.g., RGB video, audio, acceleration, infrared sequence, etc.) HAR methods have been extensively investigated in the past years Wang et al. (2016); Lin et al. (2019); García-Hernández et al. (2017); Slim et al. (2019); Akula et al. (2018). However, unimodal HAR methods limit generalization ability in real-world scenarios. The performance of a video-based HAR method is usually sensitive to illumination intensity, visual occlusion, or complex background. Meanwhile, noisy or missing data and movement variance between different users can negatively affect a sensor-based HAR method. In these cases, it is important to leverage both vision-based and sensor-based modalities to compensate for the weaknesses of single modality and improve the performance of HAR in multi-modal manners. For instance, the movement trajectories of hands are similar when people eat or drink, thus it is not easy to distinguish the two actions only based on imu sensor data. However, when the vision modality is considered, the two actions can be distinguished based on the visual characteristics of the objects held by the hands. Another example is outdoor activity recognition. Due to the similarity of the outdoor environment, it is challenging for the single-modality systems to classify whether a user is walking or running by relying only on the visual data of the smart glasses. On the other hand, the imu sensor data (i.e., accelerometer data) of these two activities reveal significantly different feature representations.

In recent years, sensors have been widely equipped in smart glasses, smartwatches, and smartphones, making the available input modalities for HAR more abundant. Therefore, various multimodal learning methods have been proposed to exploit complementary properties among modalities for fusion and co-learning-based HAR. Existing multimodal learning methods fuse vision-sensor and imu sensor data have the following shortcomings: 1) Owing to data heterogeneity, most existing methods feed uni-modal data into separate sub-networks with different structures to extract features and fuse them at the end stages. Obviously, there is a huge structure divergence between imu sensor and

vision-sensor data. Since the imu sensor data are one-dimensional time-series signals, most of the previous literature utilize 1D-CNN or LSTM network to extract spatial and temporal features of raw imu sensor data Steven Eyobu & Han (2018); Panwar et al. (2017); Wang et al. (2019a). The vision-sensor activity data, however, are usually images or videos with two or more dimensions, which is suitable for 2D-CNN or 3D-CNN to extract visual features Simonyan & Zisserman (2014); Karpathy et al. (2014b); Sun et al. (2017). Thus, the form of input of existing multi-modal learning models not only ignore the temporal synchronization correlation between multimodal data but also lose valuable complementary information. Additionally, it costs to add new sub-networks to extract the learning representations of new modality inputs. 2) Dense temporal sampling, which means sampling frames densely in a video clip or sampling the entire series of sensor data in a time period, is widely used in previous work to capture long-range temporal information in long-lasting activities. For example, those methods Tanberk et al. (2020); Wei et al. (2019) mainly rely on dense temporal sampling to improve the performance, which results in data redundancy and unnecessary computation since the adjacent frames in the video have negligible difference and the imu data of some activities are periodic. 3) Although some newly proposed attention-based multi-modal learning methods have improved the performance of HAR tasks, their complicated architectures lead to high computational overhead and make them challenging to be deployed on mobile and wearable devices. Islam & Iqbal (2021; 2022).

To address the challenges above, we propose a novel and efficient HAR multimodal network for vision and imu sensors called **MMTSA**. Our method includes three contributions:

- We design MMTSA, a novel multi-modal neural architecture based on RGB videos and IMU sensor data for end-to-end human activity recognition application. MMTSA efficiently explores information from heterogeneous multi-modal data through an isomorphism mechanism and enables extensions for new modalities. Moreover, MMTSA leverages a segment-based sparse co-sampling scheme to enable a long-lasting activity recognition at low computational resources.

- MMTSA first applies Gramian Augular Field (GAF) methods to encode the imu sensor data into multi-channel grayscale images in the multi-modal human activity recognition. The transformation arrows the structural gap among the RGB data and imu sensor data and enhances the resuability of the model's sub-networks, which bring the computation improvement.

- To better mine the complementary information between synchronized data of different modalities and the correlation between different temporal stages, we design inter-segment modality attention mechanisms for spatiotemporal feature fusion. We compared the performance of our method to several state-of-the-art HAR algorithms and traditional algorithms on three multi-modal activity datasets. MMTSA achieved an improvement of 11.13% and 2.59% (F1-score) on the MMAct dataset for the cross-subject and cross-session evaluations.

## 2 RELATED WORK

### 2.1 UNI-MODAL HUMAN ACTIVITY RECOGNITION

Single modality-based Human Activity Recognition (HAR) has been extensively investigated in the past decades. Currently, many researchers are utilizing the deep neural network-based approaches to learn single modality feature representations for HAR.

With the success of CNN He et al. (2016); Zhang et al. (2018) in image tasks, researchers put great efforts on visual modality. The accuracy of video-based HAR methods is highly related to video representation learning. Early works try to use 2D based methods Karpathy et al. (2014a) to learn video representations. Later, 3D convolutional networks Tran et al. (2015); Xie et al. (2018) are explored and achieve excellent performance. However, 3D based methods always take several consecutive frames as input, so that huge complexity and computational costs make these methods expensive to be used. Recently, 2D convolutional networks with temporal aggregation Wang et al. (2016); Lin et al. (2019) achieve significant improvements, which have much lower computational and memory

costs than 3D based methods. Although, the video-based HAR methods have good performance in most scenarios, it raises privacy concerns since visual modality contains rich appearance information of captured scene context.

Additionally, the robustness to lighting, occlusion, and subject angle are also key areas for improvement in HAR. Researchers pay attention to other single modality data, such as audio, and imu wearable sensors. However, in recent years, compared to other single modalities, using audio is not a very popular scheme for HAR. Only a few deep learning methods are proposed for HAR from audio alone Liang & Thomaz (2019). Moreover, wearable sensor data is a good choice for HAR, due to their robustness against viewpoint, occlusion and background variations, etc. Many works of literature Wang et al. (2019a) have proposed wearable sensor-based solutions for HAR. However, the accuracy of wearable sensor-based methods is sensitive to the placement on the human body Mukhopadhyay (2014). Furthermore, most of current wearable sensor-based methods perform poorly in complex HAR scenarios.

Although these single modality methods have shown promising performances in many cases, these approaches have a significant weakness, which rely on high-quality sensor data. If the single modality data is noisy and missing, the uni-modal learning methods cannot extract robust features and have poor performance in human activity recognition.

## 2.2 MULTI-MODAL HUMAN ACTIVITY RECOGNITION

In order to overcome the shortcoming of single modality missing and occlusion, multimodal learning has been used HAR. By aggregating the advantages and capabilities of various data modalities, multimodal learning can provide more robust and accurate HAR. Moreover, most existing algorithms Islam & Iqbal (2022) use each modality sensor data independently and fuse all modalities learning representations at the end of their framework for classification. Therefore, the final performance is highly related to salient feature representations of single modality. However, these architectures neglect the intrinsic synchronous property among all modalities and assume all modalities contribute to final performance equally.

To address these challenges, several works introduce new multi-modal HAR algorithms. Firstly, instead of aggregating different modalities at a late stage, TBN architecture Kazakos et al. (2019) combined three modalities (RGB, flow and audio) with mid-level fusion at each time-step and showed that visual-audio modality fusion in egocentric action recognition tasks improved the performance of both the action and accompany object. However, the mid-level fusion method is only explored in video and audio modalities and has not been extended to other sensor data (e.g. imu sensors). Secondly, attention-based approaches have recently been applied in feature learning for HAR as attention mechanism allows the feature encoder to focus on specific parts of the representation while extracting the salient features of different modalities. For example, Long et al. (2018) proposed a new kind of attention method called keyless to extract salient uni-modal features, which were combined to produce multi-modal features for video recognition. Moreover, Multi-GAT Islam & Iqbal (2021) explored the possibilities of using graphical attention methods for multi-modal representation learning in HAR.

Although these multi-modal HAR methods have achieved good performance in various scenarios, several challenges still remain in multi-modal HAR. For example, many of these methods encode the whole sensor data, which is redundant and highly computational. Therefore, a sparse and efficient sampling strategy would be more favorable and need to be designed. Furthermore, many existing frameworks do not allow inter-modality interaction and may not learn complementary multi-modal features. Thus, we explore the inter-modality and inter-segment attention mechanism and demonstrate that it improves the final result. Finally, we propose a novel multimodal temporal segment attention network MMTSA, as described in detail in the next section.

## 3 METHOD

In this section, we present our proposed method: MMTSA, a multi-modal temporal segment attention network as shown in Fig. 1. MMTSA consists of three sequential learning modules: 1) **Multimodal data isomorphism mechanism based on imu data imaging**: The module is responsible for transforming imu sensor data into multi-channel grayscale images via Gramian Angular

Field (GAF), making visual-sensor data and imu sensor data representations to be isomorphic. 2) **Segment-based multimodal sparse sampling**: We propose a novel multimodal sparse sampling strategy in this module. It performs segmentation and random sampling on RGB frame sequences and GAF images of IMU data, preserving the modal timing correlation while effectively reducing data redundancy 3) **Inter-segment attention for multimodal fusing**: To better mine the spatiotemporal correlation and complementary information between modalities, we propose an efficient inter-segment attention method to fuse multimodal features, which improve HAR performance.We discuss how MMTSA works in greater detail.

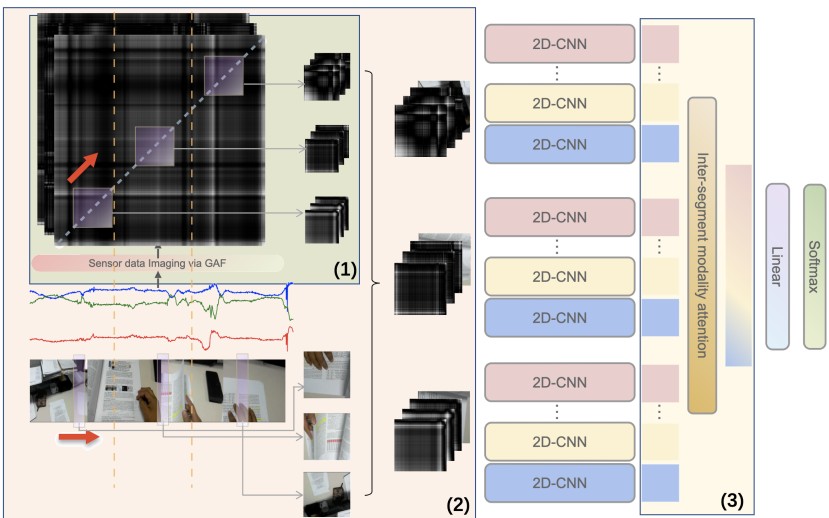

Figure 1: The architecture of MMTSA: 1) Multimodal data isomorphism mechanism. 2) Segment-based multimodal sparse sampling. 3)Inter-segment attention modality fusing.

### 3.1 MULTIMODAL DATA ISOMORPHISM MECHANISM

Although deep learning has achieved great success in CV and NLP, techniques of it fail to have many comparable developments for time series. Most traditional deep learning methods for time series build models based on RNN, LSTM, or 1D-CNN. However, these approaches have been proved to have limitations Pascanu et al. (2013); Wu et al. (2018).

To leverage the advanced 2D-CNNs and related techniques in computer vision, Wang & Oates (2015) first proposed a novel representation for encoding time series data as images via the Gramian Angular Fields (GAF) and hence used 2D-CNNs to improve classification and imputation performance. Since then, time series imaging methods have caught much attention. Inspired by the method proposed in Wang & Oates (2015), we note that GAF based methods have great potential to reduce structural differences in data from multiple modalities (e.g. RGB video and accelerometers). Therefore, we propose a multimodal data isomorphism mechanism based on GAF, which can enhance the representation ability of the temporal correlation of imu sensor data and improve the reusability of different modal feature extraction networks. We will now briefly describe how our multimodal data isomorphism mechanism works.

1) IMU sensor series rescaling : Let $S = \{s_{t_1}, s_{t_2}, \ldots, s_{t_n}\}$ be a time series collected by a imu sensor, where $s_{t_i} \in \mathbb{R}$ represents the sampled value at time $t_i$. $T = t_n - t_1$ represents the sampling time duration of this time series. We rescale $S$ onto $[-1, 1]$ by:

$$\tilde{s_{t_i}} = \frac{(s_{t_i} - \max(S) + (s_{t_i} - \min(S))}{\max(S) - \min(S)} \tag{1}$$

The max-min normalization step makes all values of $S$ fall in the definition domain of the *arccos* function, which satisfies the conditions for the coordinate system transformation.

2) Polar coordinate system transformation: In this step, we transform the normalized Cartesian imu sensor series into a polar coordinate system. For the time series $S$, the timestamp and the value of each sampled data point need to be considered during the coordinate transformation. Then we use an inverse cosine function to encode each data point $\tilde{s_{t_i}}$ into polar coordinate by:

$$\begin{cases} \phi_{t_i} = \arccos\left(\tilde{s_{t_i}}\right), -1 \leq \tilde{s_{t_i}} \leq 1, \tilde{s_{t_i}} \in \tilde{S} \\ r_{t_i} = \frac{t_i}{T}, t_i \in \mathbb{T} \end{cases} \tag{2}$$

where $\phi_{t_i}$ and $r_{t_i}$ represent the angle and the radius of $\tilde{s_{t_i}}$ in the polar coordinate, respectively. The encoding in equation2 has the following advantages. First, it is a composition of bijective functions as $\cos\left(\phi\right)$ is monotonic function when $\phi \in [0, \pi]$, which allows this transformation to preserve the integrity of the original data. Second, it preserves absolute temporal relations, as the area of $\phi_{t_i}$ and $\phi_{t_j}$ in polar coordinates is dependent on not only the time interval of $t_i$ and $t_j$, but also the absolute value of them Wang & Oates (2015). The coordinate transformation above maps the 1D time series into a 2D space, which is imperative for calculating the Gramian Angular Field latter.

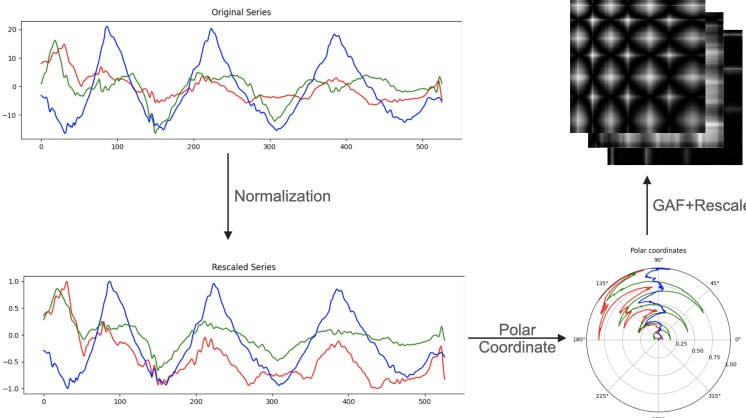

Figure 2: IMU sensor data imaging via Gramian Angular Field in MMTSA

3) Image generation via Gramian Angular Field (GAF): The Gram Matrix G is the matrix of all inner products of $X = \{x_1, x_2, \ldots, x_n\}$, where $x_i$ is a vector. A dot product in a Gram Matrix can be seen calculating similarity between two vectors. However, in the polar coordinate mentioned above, the norm of each vector causes a bias in the calculation of the inner product. Therefore, we exploit the angular perspective by considering the trigonometric sum between each point in the polar coordinate to identify the temporal correlation within different time intervals, which is an inner product-like operation solely depending on the angle. The GAF is defined as follows:

$$G(\tilde{S}) = \begin{bmatrix} \cos\left(\phi_{t_1} + \phi_{t_1}\right) & \cdots & \cos\left(\phi_{t_1} + \phi_{t_n}\right) \\ \cos\left(\phi_{t_2} + \phi_{t_1}\right) & \cdots & \cos\left(\phi_{t_2} + \phi_{t_n}\right) \\ \vdots & \ddots & \vdots \\ \cos\left(\phi_{t_n} + \phi_{t_1}\right) & \cdots & \cos\left(\phi_{t_n} + \phi_{t_n}\right) \end{bmatrix} \tag{3}$$

where

$$\cos\left(\phi_{t_i} + \phi_{t_j}\right) = \cos(\arccos(\tilde{s_{t_i}}) + \arccos(\tilde{s_{t_j}}))$$
$$= \tilde{s_{t_i}} \cdot \tilde{s_{t_j}} - \sqrt{1 - \tilde{s_{t_i}}^2} \cdot \sqrt{1 - \tilde{s_{t_j}}^2}, 1 \leq i, j \leq n \tag{4}$$

The GAF representation in a polar coordinate maintains the relationship with the original time-series data via exact inverse operations. Moreover, the time dimension is encoded into GAF since time increases as position moves from top-left to bottom right, preserving temporal dependencies.

Then we map each element in the GAF representation to a pixel of a grayscale image by:

$$\tilde{G}_{i,j} = \frac{\cos(\phi_{t_i} + \phi_{t_j}) - \min(\tilde{G})}{\max(\tilde{G}) - \min(\tilde{G})} \times 256, 1 \leq i, j \leq n \tag{5}$$

where $\tilde{G}$ is a grayscale image of size $n \times n$.

Most wearable sensors (accelerometers, gyroscopes, magnetometers, etc.) are triaxial. Suppose given a time series data $\{S^{(x)}, S^{(y)}, S^{(z)}\}$ sampled by a 3-axis accelerometer, we can generate three grayscale images $\{\tilde{G}^{(x)}, \tilde{G}^{(y)}, \tilde{G}^{(z)}\}$ for the $x$, $y$, and $z$ axes according to the above steps. After concatenating these three images, the time series sampled by each three-axis imu sensor can be uniquely converted to a multi-channel grayscale image of size $3 \times n \times n$.

## 3.2 SEGMENT-BASED MULTIMODAL SPARSE SAMPLING

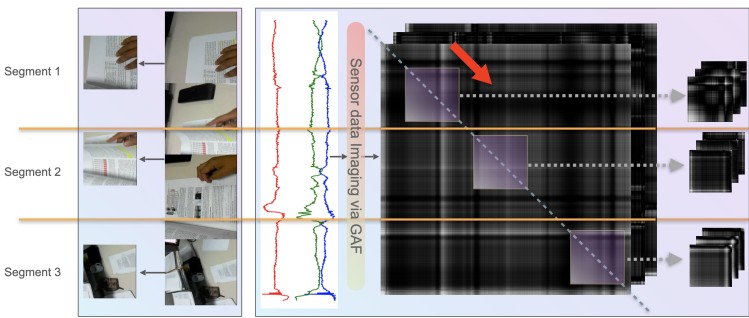

Figure 3: Multimodal sparse sampling strategy in MMTSA

The two mainstream methods of traditional human activity recognition based on video understanding are 3D-CNN and two-stream CNN networks, but the limitations of these two schemes are that they only can capture short-range temporal dependencies in the video. In order to capture long-distance temporal dependencies, these methods usually require densely sampled video clips. A video clip is $m$ consecutive frames sampled by a sliding window of size $m$ in a period, and the whole video is divided into several clips. However, the content changes relatively slowly between two adjacent frames in a clip, which means the sampled consecutive frames are highly redundant.

Similar to visual data, we observe redundancy in imu sensor data. As shown in Fig.2, the imu sensor data collected from the 3-axis accelerometer located on the smartwatch has obvious periodicity while the user was waving hands. We reasoned that this phenomenon should be common in imu sensor data. In most daily activities, human limb movements are regular and repetitive, so there is a corresponding periodicity in the data collected by wearable sensors. It means that local data features can represent the characteristics of the entire activity, thus the complete time series is redundant. However, traditional deep learning activity recognition models based on imu sensor data and newly proposed ones ignored this issue. In previous work, a common input method is to feed the complete collected data series into deep models Xu et al. (2021); Wang et al. (2019b), such as CNN, RNN, LSTM, or Bert. Another widely-used method is dense sampling in fixed-width sliding windows and overlap Bianchi et al. (2019). These unnecessary dense sampling methods lead to larger memory overhead and longer inference time. In addition, the above dense sampling strategy will consume too much energy when deployed on the device.

Wang et al. (2016) proposed TSN framework to deal with the frame redundancy in video understanding by applying a sparse and global temporal sampling strategy. This strategy divides the video into a fixed number of segments and one snippet is randomly sampled from each segment. To overcome the aforementioned challenges of data redundancy in multimodal tasks, we leverage the segmentation idea of Wang et al. (2016) and propose a sparse sampling method for multi-wearable sensor time series, as shown in Fig.3.

We find that the multi-channel grayscale images generated by GAF based on imu sensor data have some excellent properties. First, the diagonal of each grayscale image is made of the original value of the scaled time series ($G_{i,i} = 2\tilde{s}_{t_i}^2 - 1$). Second, the sampling along the diagonal direction has local temporal independence. Given two timestamps $t_i$ and $t_j$ ($i \leq j$), we sample a square area in the grayscale image with a diagonal extending from $\tilde{G}_{i,i}$ to $\tilde{G}_{j,j}$. The data in this square matrix only depends on the timestamps between $t_i$ and $t_j$, and represents the temporal correlation of the original series in this period. Our proposed method firstly divides the entire imu sensor data

into N segments of equal duration according to timestamps. The dividing points of these segments correspond to equidistant points on the diagonal of the grayscale image generated based on GAF: $\{\tilde{G}_{(S_0,S_0)}, \tilde{G}_{(S_1,S_1)}, \cdots, \tilde{G}_{(S_N,S_N)}\}$. In each segment, we use a square window of size $K$ for random multi-channel sampling:

$$\tilde{G}(S_i) = \begin{bmatrix} \tilde{G}_{(i,i)} & \cdots & \tilde{G}_{(i,i+K-1)} \\ \vdots & \ddots & \vdots \\ \tilde{G}_{(i+K-1,i)} & \cdots & \tilde{G}_{(i+K-1,i+K-1)} \end{bmatrix} \tag{6}$$

where $S_{i-1} \leq i < i+K-1 \leq S_i$. For multi-axis sensors, random sampling of each segment is performed simultaneously on multiple channels.

### 3.3 INTER-SEGMENT AND MODALITY ATTENTION MECHANISM

To fuse features of different modal data in each segment and extract more spatiotemporal information in multimodal training, we propose an additive attention-based inter-segment modality fusion mechanism in MMTSA, which is effective and efficient. We first concatenate the features of different modalities in each segment by:

$$Y_{S_i} = \textbf{Concat}\{\mathcal{F}\left(X^1_{S_i}; \mathbf{W^1}\right), \mathcal{F}\left(X^2_{S_i}; \mathbf{W^2}\right), \cdots, \mathcal{F}\left(X^m_{S_i}; \mathbf{W^m}\right)\} \tag{7}$$

Where $Y_{S_i}$ is the output of the $i$-th segment, $\mathcal{F}\left(X^j_{S_i}; \mathbf{W^j}\right)$ represents a ConvNet with parameters $\mathbf{W^j}$ that operates on $X^j_{S_i}$ and $j$ indicates the $j$-th modality. $X^j_{S_i}$ and $\mathbf{W^j}$ represent the sampled data of the $j$-th modality in segment $i$ and the shared parameters of modality $j$, respectively. Then we get an output sequence of each segments: $Y^{out} = (Y_{S_1}, Y_{S_2}, \cdots, Y_{S_N})$ where $Y_{S_i} \in R^{M \cdot d}$ and $d$ is the feature dimension of each modality.

We utilize additive attention to calculate the attention weight of each segment,

$$\alpha_{S_i} = \frac{\exp\left((W^{att})^T Y_{S_i}\right)}{\sum_{i \in N} \exp\left((W^{att})^T Y_{S_i}\right)} \tag{8}$$

where $W^{att} \in R^{M \cdot d}$ is a learnable parameter and $(W^{att})^T Y_{S_i}$ is a score to evaluate the importance of each segment. The softmax function is used to regularize the scores so that the sum of the scores of all segments is 1. Here, the parameter $W^{att}$ will weight the features of different modalities to compensate for the inter-modal information.

Next, the attention weights $\alpha_S = \{\alpha_{S_1}, \alpha_{S_2}, \cdots, \alpha_{S_N}\}$ will be used to fuse the outputs of each segment and get a weighted global representation,

$$\tilde{Y}_{S_i} = \sum_{i \in N} \alpha_{S_i} Y_{S_i} \tag{9}$$

Finally, the global representation $\tilde{Y}_{S_i}$ is fed into a Feed Forward Neural Network with two fully-connected layers and a softmax to compute the probabilities for each class of human activities.

## 4 EXPERIMENT

### 4.1 DATASET

We evaluate our proposed model multimodal temporal segment attention network MMTSA for the human activity recognition task on three public datasets: Multimodal Egocentric Activity Song et al. (2014), DataEgo Possas et al. (2018), and MMAct Kong et al. (2019).

### 4.2 EXPERIMENT SETTINGS

We compare our proposed model multimodal temporal segment attention network with the following state-of-the-art multimodal HAR training algorithms for comparison, such as TSN Wang et al.

Table 1: cross-subject and cross-session performance comparison on MMAct dataset

| Cross-Suject | F1-Score (%) | Cross-Session | F1-Score (%) |
|---|---|---|---|
| SMD Hinton et al. (2015) | 63.89 | SVM+HOG Ofli et al. (2013) | 46.52 |
| Multi-Teachers Kong et al. (2019) | 62.27 | TSN (Fusion) Wang et al. (2016) | 77.09 |
| Student Kong et al. (2019) | 64.44 | MMAD Kong et al. (2019) | 78.82 |
| MMAD Kong et al. (2019) | 66.45 | Keyless Long et al. (2018) | 81.11 |
| Keyless Long et al. (2018) | 71.83 | HAMLET Islam & Iqbal (2020) | 83.89 |
| Multi-GAT Islam & Iqbal (2021) | 75.24 | MuMu Islam & Iqbal (2022) | 87.50 |
| MuMu Islam & Iqbal (2022) | 76.28 | Multi-GAT Islam & Iqbal (2021) | 91.48 |
| **MMTSA (our method)** | **87.41** | **MMTSA (our method)** | **94.07** |

(2016), Keyless Long et al. (2018), HAMLET Islam & Iqbal (2020), MuMu Islam & Iqbal (2022). We use the mirco F1 score to evaluate the performance of all methods.

**Data preprocessing**: The imu data of each sensor is imaged based on the method in Sec.3.1. To achieve the efficiency of our method, we directly segment the imu series and randomly select a timestamp within each segment. The window containing the timestamp is used to sample the original series, which is equivalent to strategies in Sec.3.2. The RGB video and its synchronous imu data series is divided into the same number of segments, and a frame is randomly sampled in each segment.

**Training Details:** We implemented our proposed method in Pytorch. We utilized Inception with Batch Normalization (BN-Inception) as a sub CNN to extract the unimodal features learning representations. Moreover, in the proposed model, we trained all the modalities simultaneously with $N = 3$ segments, SGD with momentum optimizer, and a learning rate of $0.001$. The convolutional weights for each modality are shared over the $N$ segments, and hence reduce the model-size and memory cost.

## 5 RESULT

### 5.1 RESULTS COMPARISON

We evaluate our proposed model MMTSA performance and summarize all the results. For MMAct dataset, we follow originally proposed cross-subject and cross-session evaluation settings and report the results in Table 1. The results show that MMTSA improves $11.13\%$ and $2.59\%$ in cross-subject and cross-session evaluation settings Kong et al. (2019). For Multimodal Egocentric Activity, we follow leave-one-subject-out cross validation. MMTSA outperforms all of the traditional methods and is close to the performance of MFV as shown in Table 2. Compared to MFV that uses four types of sensor data, MMTSA only uses two types (accelerometer, gyroscope) as input to make the model lightweight. Thus, a small loss of precision is acceptable. For DataEgo, we divide each 5-minute original data into 15 seconds clips with 5-second overlapping. We keep the train and test split size of each activity balance. We show the cross-subject performance in Table 3, which outperforms TSN by $17.45\%$.

Table 2: cross-subject performance comparison on Multimodal Egocentric Activity

| Method | F1-Score (%) |
|---|---|
| SVM Kwapisz et al. (2011) | 47.75 |
| Decision Tree Kwapisz et al. (2011) | 51.80 |
| FVS Song et al. (2016b) | 65.60 |
| TFVS Song et al. (2016b) | 69.00 |
| Multi-Stream with average pooling Song et al. (2016a) | 76.50 |
| FVV Song et al. (2016b) | 78.44 |
| FVV+FVS Song et al. (2016b) | 80.45 |
| Multi-Stream with maximum pooling Song et al. (2016a) | 80.50 |
| **MMTSA (our method)** | **81.00** |
| MFV Song et al. (2016b) | 83.71 |

Table 3: cross-subject performance comparison on DataEgo

| Method | F1-Score (%) |
|---|---|
| TSN (RGB) Wang et al. (2016) | 65.77 |
| **MMTSA (our method)** | **83.22** |

## 5.2 ABLATION STUDY

This section is organized as follows. First, we show and discuss the performance of single modalities, and compare them with our proposed model MMTSA. Second, we compare direct feature concatenation and our additive inter-segment attention in feature fusion stage. Specifically, in this section, all the experiments are done on the MMact dataset and followed the original proposed cross-subject evaluation settings.

**Single Modality vs Multi-Modality fusion:** We compare the performance of a single modality and different combinations of modalities in Table 4. The fusion of all modalities obtains the best results, which indicates that the multimodal isomorphism and fusion mechanism in MMTSA mines the complementary information among modalities more comprehensively.

**Effectiveness of inter-segment attention modality fusion:** We compare the simple concatenation method and our proposed inter-segment attention fusion method. The results shown in 5 suggest that our proposed method outperforms the concatenation method. It shows that the inter-segment attention modality fusion method has a significant effect on the fusion of spatiotemporal features between modalities. Moreover, during training, we also find that the use of inter-segment attention leads to a faster convergence speed.

Table 4: modality ablation experiment on MMAct (cross-subject)

| Modality | F1-Score (%) |
|---|---|
| RGB | 66.95 |
| AccWatch+AccPhone+Gyro+Orie | 82.13 |
| RGB+AccWatch+AccPhone | 84.47 |
| **RGB+AccWatch+AccPhone+Gyro+Orie** | **87.41** |

Table 5: fusion method experiment on MMAct (cross-subject)

| Fusion method | F1-Score (%) |
|---|---|
| Concat | 84.02 |
| **Inter-segment attention** | **87.41** |

## 6 CONCLUSION

In this work, we presented MMTSA, a new novel multi-modal neural architecture based on RGB and IMU wearable sensors for human activity recognition. MMTSA employs a multimodal data isomorphism mechanism by encoding the IMU sensor series into GAF images, which potentially reduces the structure differences of multiple modalities and enhances the reusability of the model's subnetworks. Additionally, our proposed segment-based sparse sampling method on multimodal input data is efficient and reduce data redundancy. Moreover, by adding the additive inter-segment attention mechanism, our model enabled features interactions among different modalities and achieved better final results. As demonstrated on three public datasets, Multimodal Egocentric Activity, DataEgo, and MMAct, our method significantly improved state-of-the-art performance. Furthermore, in the future, we plan to deploy our proposed model MMTSA on device to enable safe and accurate human activity recognition in real-world setting.

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
