# OpenReview forum: "MMTSA: Multi-Modal Temporal Segment Attention Network for Efficient Human Activity Recognition"
_ICLR.cc/2023/Conference — Submitted to ICLR 2023_

### Official Review · Reviewer_6cv1 · 2022-10-23

**Confidence:** 4
**Correctness:** 4
**Technical Novelty And Significance:** 2
**Empirical Novelty And Significance:** 3
**Recommendation:** 5

**Clarity, Quality, Novelty And Reproducibility:**

**Clarity**
1. I would recommend adding some qualitative results (figures of activity samples correctly and incorrectly classified, t-SNE plots) to add more perspective to the network performance.

2. A list of "limitations" of the proposed method will be helpful to better understand its contributions, especially w.r.t. noise robustness and the modalities it has been tested with.


**Quality and Novelty**

The paper appears to be a combination of known techniques for the most part. While the experimental results show appreciable improvements, the paper can benefit from a description of the technical limitations and challenges that led to the design.


**Reproducibility**

The authors have clearly explained the components of their method and they are reproducible.

**Strength And Weaknesses:**

**Strengths**
1. The proposed approach of transforming the IMU sensor data to grayscale images is sound and interesting. The transformation process is also clearly explained.
2. The proposed attention mechanism provides appreciable performance improvements as per the authors' experiments.
3. The paper is overall well-written and well-organized.


**Weaknesses**
1. While the paper nicely explains the different components of the proposed method, those components all appear to be based on prior work. I am not clear on the technical limitations this work overcomes compared to prior work. For example, what are the limitations in processing the 1D IMU sensor data using simpler recurrent networks and then combining the downstream features? How does "converging" the structural differences between the RGB and the IMU sensor data help with the temporal correlation that the authors mention in Sec. 1? Related to this question, are there any supporting works (or experiments performed by the authors) that show how and why existing multimodal methods ignore the temporal correlations between these types of data? On the other hand, why have the authors not considered optical flow alongside RGB video data, given that optical flow methods are popular for video-based activity recognition (such as Tanberk et al. 2020 and the seminal work of Simonyan and Zisserman 2014).

2. The authors mention working with mobile devices and wearables, where the data can be highly noisy (shaky camera, general arm movements not related to any particular activities). How do the authors deal with the noise in their end-to-end pipeline? A simple experiment to demonstrate noise robustness is to artificially add noise into cleaned data and monitor the network performance as the noise level is increased.

3. Is the proposed attention mechanism specifically designed for the RGB and the IMU sensor modalities or can it be reused and extended to other modalities such as audio and physiological signals such as heart rate?

**Summary Of The Paper:**

The authors present a method to perform human activity recognition using a novel multimodal temporal segment attention network leveraging a combination of RGB videos and IMU sensor (accelerometer and gyroscope) data from wearable sensors. To bring the IMU sensor data closer to the RGB data, the authors convert them to grayscale images using a Gramina Angular Field-based isomorphism. They perform sparse co-sampling of both the RGB and the transformed IMU sensor images to reduce the temporal redundancy and design attention networks to eventually predict the activity labels. The authors perform experiments to demonstrate the benefits of using their multimodal data and their attention mechanism.

**Summary Of The Review:**

The paper presents an interesting method for human activity recognition using a multimodal attention mechanism on RGB video data and structurally similar grayscale images of transformed IMU sensor data. However, the components of the proposed method are all built on prior work and I am not clear on the technical limitations that the authors overcame by coming up with the design of their method. I invite the authors to respond to my concerns under "weaknesses" to get a clearer picture of their contributions before I can recommend acceptance.

---

> ### Author Response · Authors · 2022-11-19
> **Response to Reviewer 6cv1 [1/2]**
>
> Thank you for your interest and comments. We answer your questions one by one below.
>
> > "what are the limitations in processing the 1D IMU sensor data using simpler recurrent networks and then combining the downstream features? "
>
> In the previous work [1-2], processing the 1D IMU sensor data based on RNN, LSTM, or 1D-CNN has been proven to have limitations. Firstly, it is difficult to train RNN due to vanishing and exploding gradients. Secondly, there are already extensive experiments that show that 2D-CNN methods perform better than 1D-CNN methods for time series feature extraction. Moreover, to achieve better performance, these methods rely on sampling the entire series of sensor data in a time period, which is redundant and unnecessary, especially in periodic activities. In MMTSA, we convert the IMU sensor data into images via GAF and apply a sparse sampling method, which is effective and efficient.
>
> > "How does 'converging' the structural differences between the RGB and the IMU sensor data help with the temporal correlation that the authors mention in Sec. 1"
>
> The isomorphism of IMU data and RGB data is to increase the sub-network's reusability, reduce the model's complexity, and extract features more effectively. First, when other one-dimensional modal data are added to the input, we can transform it into two-dimensional data based on GAF and directly reuse the MMTSA sub-network to extract features. Second, the GAF-based modality isomorphism method ensures that the feature dimensions of different modalities are the same, facilitating feature fusion at the segment level. Third, based on the properties of GAF, converting one-dimensional data into two-dimensional images can better extract spatiotemporal features.
>
> > "Are there any supporting works (or experiments performed by the authors) that show how and why existing multimodal methods ignore the temporal correlations between these types of data? "
>
> In [3], the author performed a GAF-based transformation on the IMU data, distilling the trained model's knowledge into the model corresponding to the RGB data. The author proves through experiments that the one-dimensional modal data can be better complemented with the two-dimensional modal data after GAF imaging because the GAF-based transformation can extract more temporal information. Experiments also show that this GAF-based knowledge distillation method outperforms other methods.
>
> >"Why have the authors not considered optical flow alongside RGB video data?"
>
> Because we plan to deploy our model on the device, processing the RGB video data into optical flow obviously increases modal size and memory cost, even though the optical methods are popular.
>
> > "How do the authors deal with the noise in their end-to-end pipeline, where the data can be high noise (shaky camera, general arm movements not related to any particular activities)?"
>
> In our current submission, we do not perform the noise experiment in our end-to-end pipeline for the following reasons. Firstly, we evaluate our proposed model MMTSA on three public datasets: Multimodal Egocentric Activity [4], DataEgo [5], and MMAct [6]. Take the dataset DataEgo for example, the egocentric video data are collected in a real scenario. Especially in some activities, the egocentric video data are motion blurred, and the light changes. Secondly, by aggregating information from different modalities and applying the attention mechanism in our model, the noise in the single modality problem is reduced to some degree. In the future, we plan to include the experiment to demonstrate noise robustness and model performance as the noise level increases.
>
> > "Is the proposed attention mechanism specifically designed for the RGB and the IMU sensor modalities or can it be reused and extended to other modalities such as audio and physiological signals such as heart rate?"
>
> The proposed attention mechanism is not specifically designed for the RGB and the IMU sensor modalities and it can be reused and extended to other modalities such as audio and physiological signals such as heart rate. In our model MMTSA, we apply an attention mechanism to learn the weights of combining different modalities learning representations in each segment and learn the weights of combining each segment, since different modalities may focus on different time periods. The attention mechanism enables our model MMTSA to aggregate spatiotemporal information effectively and efficiently. Moreover, the audio data can be preprocessed into images via the log-spectrogram representation method, and physiological signals are 1D time series data, which can be converted into images via GAF methods as we mentioned in our paper. Thus our model MMTSA can be easily extended to deal with other modalities.

---

> > ### Author Response · Authors · 2022-12-02
> > **Response to Reviewer 6cv1 [2/2]**
> >
> >  **Reference**
> >
> > [1] Razvan Pascanu, Tomas Mikolov, and Yoshua Bengio. On the difficulty of training recurrent neural networks.
> >
> > [2] Yunan Wu, Feng Yang, Ying Liu, Xuefan Zha, and Shaofeng Yuan. A comparison of 1-d and 2-d deep convolutional neural networks in ecg classification.
> >
> > [3] Liu, Yang, et al. "Semantics-aware adaptive knowledge distillation for sensor-to-vision action recognition.
> >
> > [4] Sibo Song, Vijay Chandrasekhar, Ngai-Man Cheung, Sanath Narayan, Liyuan Li, and Joo-Hwee Lim. Activity recognition in egocentric life-logging videos.
> >
> > [5] Rafael Possas, Sheila Pinto Caceres, and Fabio Ramos. Egocentric activity recognition on a budget.
> >
> > [6] Quan Kong, Ziming Wu, Ziwei Deng, Martin Klinkigt, Bin Tong, and Tomokazu Murakami. Mmact: A large-scale dataset for cross modal human action understanding.

---

### Official Review · Reviewer_hEtV · 2022-10-24

**Confidence:** 4
**Correctness:** 3
**Technical Novelty And Significance:** 1
**Empirical Novelty And Significance:** 2
**Recommendation:** 3

**Clarity, Quality, Novelty And Reproducibility:**

Clarity - the paper is generally very clear. There are a couple of places which could have expanded a bit more, e.g. the end of Sec 3.2 is very sparse on how the random segments are created.
Quality - in general the paper is of good quality. It describes the approach well and provides good details around prior work usage and experiments.
Novelty - the approach presented in the paper is not very novel and the paper is mainly a basic application paper that does perform better than some state-of-the-art but does not provide any new insights. Also see comments above.
Reproducibility - the authors have provided a good amount of detail that would be helpful towards experimenting with the broad approach. Not sure whether the exact results would be reproducible without knowing all the values of the parameters used.


**Strength And Weaknesses:**

Strengths:

+ The approach is fairly well described and appears to be a reasonable application of attention to segments of GAF images.
+ The results do show mild improvements above the state-of-the-art.

Weaknesses:
- The main weakness here is the lack of novelty in the paper. The paper takes about two pages just to describe GAFs which is previously published work by Wong and Oates (2015). To be clear, the paper cites their work correctly, however this seems to be the main technology used in the paper in addition to learned attention weighting of the different modalities. Attention mechanisms and transformers are also fairly well known works in the field. Therefore, this paper is more of an application paper than one that proposes novel ideas.
- Given that the novelty in the paper is low, I would look for deep analysis on insights learned from the application to the datasets provided. The paper does provide results showing improvements with their proposed network but does not go deep into understanding why. Also there are no discussions on error modes or where the method works better or worse than state-of-the-art.
- The authors proposed using attention for weighing multiple segment features across a large sequence. However the way the features were constructed using GAF, there are already correlations computed across longer time segment ranges. Why not build a feature with these correlations as opposed to using only the diagonals of the GAF? This kind of deeper analysis would have been helpful.

**Summary Of The Paper:**

This paper proposes a method for multi-modal human activity recognition using RGB and IMU data. The main contributions include 1) modeling IMU data using several 2D images following the approach of Wang and Oats (2015) to build a Gramian Angular Field (GAF). The paper then proposes 2) extracting equal sized diagonal matrices from the GAF and then creates another multi-channel image that randomly subsamples this image several times to create a richer representation. This is the IMU feature representation multiscale input image that is then 3) combined with the RGB subscaled images through a learned attention mechanism to create a final representation that through a few NN layers and softmax returns the probability of each human activity class.

The paper provides results on three public datasets and provides comparable or better results than state-of-the-art. The paper also provides some ablation studies demonstrating better results with multi-modality using all the signals compared to other combinations with fewer signals and also demonstrates the benefits of using the attention module above just using concatenated features.

**Summary Of The Review:**

As mentioned above, the main concerns I have are around minimal novelty and no deep insights into why the approach performs better than the previous works. The paper does have an ablation study showing attention improvements against simple concatenation. However as mentioned above there are some loose ends (why not use the correlations in the GAF directly) that reduce the impact of the paper's experiments. I would have loved to see more experimentation, demonstration of improvements on specific examples against state of the art and a discussion for what may be missing for the results that are still not performing well enough. In my opinion, this paper requires more empirical work as well as a discussion that could provide deeper insights into the area. It is not ready for publication.

---

### Official Review · Reviewer_4Euk · 2022-10-24

**Confidence:** 3
**Correctness:** 2
**Technical Novelty And Significance:** 3
**Empirical Novelty And Significance:** 2
**Recommendation:** 6

**Clarity, Quality, Novelty And Reproducibility:**

The paper is well written and easy to read. The state of the art is correctly covered. The overall coverage of the method is adequate. Personally, I’d rather see less details about how to generate the GAD images and further details on how to synchronize the data and the attention mechanism

The novelty is limited but, as I stated previously, I believe it’s enough. I think the contribution of bringing together different modalities into a single architecture, via transforming IMU signals as images, can be of interest for the audience.

The experimental setup can also be improved. I’d appreciate more details regarding the data: distribution, labels, size, subjects etc

Minor & style
- Figure 1 - Illustrate which part of the illustration corresponds to each element of the architecture
- Typo:  dimension -> dimension “ Moreover, the time dimention is encoded into GA”
-Typo: “the mirco F1”
- “The two mainstream methods of traditional human activity recognition based on video understanding are 3D-CNN and two-stream CNN networks, but the limitations of these two schemes” I’d suggest to support that statement with proper references


**Strength And Weaknesses:**

This work indeed includes quite some strengths. This work presents an approach which I consider can be of interest for the community. It explores the representation of IMU signals as images, modeled by vision based approaches, completed with RGB data. The overall novelty of the work is not too strong, since it combines already known data representation with also known modeling on not novel datasets, however the combination of all I believe brings an interesting proposal.

The ablation study, although not extensive, I also think is quite interesting since it helps to illustrate the potential contribution of each data modality.

However, the paper also presents some weaknesses. In my opinion the main flaw of the paper is the discrepancy between the technical challenges the authors claim to overcome and the final results and discussion.

- An interesting point the authors raise regarding existing architectures is how “ … attention-based multi-modal learning methods [have] complicated architectures lead to high computational overhead and make them challenging to be deployed on mobile and wearable devices”, which I totally share. However I do not see how the proposed model simplifies such state of art to any degree. The approach is not characterized nor benchmarked. Results are not aimed toward measuring performance. I’d love to see how this proposal can bring a computation improvement and can work towards a feasible model running in a wearable device. Have the authors included any data in that regard that I may have missed?

- Another claim is how this approach leverages the synchronous property among data modalities, however for me it’s not clear at all how this approach enables the synchronization of data from different modalities (which indeed need to be synced in order to be used by MMTSA). Can authors explain or maybe add further information regarding this point?

I also believe that this work would greatly benefit from a deeper discussion, to come up as a much stronger work.
- Some explanation about how the data synchronization among modalities i feasible
- Some covering on the challenges of data labeling for this domain, which potentially may require to label video and sensor data. What are the options for getting training data? Would it be possible to train an independent network on independent datasets? Or finetune iteratively?
- A more clear statement (and results) on how the attention mechanism is contributing to the model, is the key for syncing data?
- How the model is simplifying (if any) current state of art



**Summary Of The Paper:**

Authors propose the MMTSA architecture, which stands for “Multimodal Temporal Segment Attention Network for Efficient Human Activity Recognition”. MMTSA is a network aimed to perform sensor fusion for imu and video data. By transforming the sensor data as GAF images, sampling them and using an attention mechanism, authors show an efficient approach for merging multi-modality data, modeling it as video data. The approach is tested on three public datasets, showing significant improvements and the overall feasibility of this architecture.

**Summary Of The Review:**

The paper presents an interesting approach which I believe can be of potential interest for practitioners. However it would need to cover better the claims and initial statements to increase the value of the work. I think this work would benefit greatly from a more extensive and deeper discussion. I’d rather see less theory about the GAF images and more details about data syncing (the attention mechanism?), real life challenges when dealing with multimodal data,  and more insights regarding the experiments.

---

> ### Author Response · Authors · 2022-11-19
> **Response to Reviewer 4Euk [1/2]**
>
> We thank Reviewer 4Euk for the constructive reviews. We were encouraged to hear that Reviewer 4Euk found the architecture of MMTSA brought an interesting proposal. We addressed the questions and concerns of the reviewer accordingly in the following.
> > "Compared with state-of-the-art methods, how does the proposed model MMTSA simplify the previous methods, bring a computation improvement and run in a wearable device?"
>
> We thank Reviewer 4Euk for noting the computational cost of MMTSA. We calculated the model size of state-of-the-art methods and MMTSA, which are listed in the table below.
> |  Methods  | Number of Parameters |
> |  ----  | ----  |
> | HAMLET [2]   | 24.90M |
> | MuMu [1]  | 25.08M |
> | Multi-GAT [3]   | 37 M |
> | **MMTSA (our method)**  | **6.48 M** |
>
> The number of parameters of our proposed approach, MMTSA, with acceleration, gyroscope, orientation and RGB videos modalities is _6.48 M_, much smaller than other SOTA methods. Moreover, MMTSA outperforms other SOTA methods on MMAct Dataset (Section 5.1).
> We have already demonstrated MMTSA can be deployed on Raspberry Pi 4B and each batch of size 1 with data from five modalities takes approximately _3.51s_.
>
> The SOTA methods we compared encode each modality independently and apply hierarchical attention modules to fuse all learning features from different modalities, increasing model parameters. For example, Mumu uses ResNet50 and Co-occurrence approach [4] to extract the image and IMU features, respectively. It then applies an attention-based fusion method in the two-stage classification. The complex architecture of Mumu leads to high computational costs. However, MMTSA uses BN-Inception as a sub-CNN to extract the representations of IMU signals and RGB data and train all modalities simultaneously, which is memory-intensive. Moreover, BN-Inception offers a good compromise between performance and model size for our model MMTSA training. Our inter-segment modality attention mechanism can improve model performance further with less computation cost.
>
> > "It’s not clear at all how this approach enables the synchronization of data from different modalities (which indeed need to be synced in order to be used by MMTSA)."
>
> > "Some explanation about how the data synchronization among modalities is feasible."
>
> >  "Some covering on the challenges of data labeling for this domain, which potentially may require to label video and sensor data. What are the options for getting training data? "
>
> We thank Reviewer 4Euk for noting the data synchronization in our approach. We evaluated MMTSA on three public datasets: Multimodal Egocentric Activity [5], DataEgo [6], and MMAct [7]. In these three public datasets, the data of different modalities are collected synchronously. Details about the data collection process, equipment, labeling methods are presented in [5,6,7].  Moreover, we think that our main research problem is not the synchronization of multimodal data, but how to better fuse multimodal features and perform well on HAR tasks with low computational cost. Since we already have the synchronous multimodal data, MMTSA employs the synchronous property at the sparse sampling stage to reduce the model size and redundant data. The  synchronous property also supports the intre-segment attention mechanism to weigh importance of different segments. Furthermore, in our future work we will explore our proposed architecture to deal with the asynchronous data by performing channel exchange during the attention-based fusion.
>
> We have considerated the synchronization challenges in multimodal data collection and now we are working on our own HAR dataset, which will be open sourced in the future. The volunteers with head-mounted AR glasses perform natural behaviors, which can capture the egocentric video and imu information at the same time. Meanwhile, all the data is transferred to the mobile phone. The APP we developed will synchronize the data of different devices and modalities according to the time of the mobile phone. The scarcity of labeled data is a common challenge in HAR. For synchronized multimodal data, we can annotate only the visual data. In our experiments.
>
> >  "Would it be possible to train an independent network on independent datasets, and finetune iteratively?"
>
> Currently, MMTSA utilizes BN-Inception as a base-architecture to extract the RGB data and IMU sensor data learning representations and trains all the modalities simultaneously end-to-end without a pretrained model. In the future, we will explore the robustness of MMTSA by using a pretrained model and fine-tuning it in few-shot cases.

---

> > ### Author Response · Authors · 2022-11-19
> > **Response to Reviewer 4Euk [2/2]**
> >
> > > "How the attention mechanism is contributing to the model, and the attention mechanism is the key for synchronizing the data?"
> >
> > In MMTSA, the attention mechanism is not the key for synchronizing the data. However, the attention mechanism is designed for better fusing multimodal features and improving the model performance. The ablation study in our paper has shown that the inter-segment attention mechanism has a significant effect on recognition performance (Table 5). In detail, the additive attention-based inter-segment modality fusion mechanism allows MMTSA to learn the importance of different modality in each segment, and simultaneously weigh the different segments. Therefore, MMTSA can capture the spatio-temporal information from different modalities effectively and efficiently via the inter-segment attention mechanism (Section 3.3).
> >
> >
> > **References**
> >
> > [1] Md Mofijul Islam and Tariq Iqbal. Mumu: Cooperative multitask learning-based guided multimodal fusion
> >
> > [2] Md Mofijul Islam and Tariq Iqbal. Hamlet: A hierarchical multimodal attention-based human activity recognition algorithm
> >
> > [3] Md Mofijul Islam and Tariq Iqbal. Multi-gat: A graphical attention-based hierarchical multimodal representation learning approach for human activity recognition
> >
> > [4] Li, C.; Zhong, Q.; Xie, D.; and Pu, S. 2018. Co-occurrence feature learning from skeleton data for action recognition and detection with hierarchical aggregation
> >
> > [5] Sibo Song, Vijay Chandrasekhar, Ngai-Man Cheung, Sanath Narayan, Liyuan Li, and Joo-Hwee Lim. Activity recognition in egocentric life-logging videos
> >
> > [6] Rafael Possas, Sheila Pinto Caceres, and Fabio Ramos. Egocentric activity recognition on a budget
> >
> > [7] Quan Kong, Ziming Wu, Ziwei Deng, Martin Klinkigt, Bin Tong, and Tomokazu Murakami. Mmact: A large-scale dataset for cross modal human action understanding

---

### Official Review · Reviewer_9pPc · 2022-10-25

**Confidence:** 5
**Clarity, Quality, Novelty And Reproducibility:** See above
**Correctness:** 3
**Technical Novelty And Significance:** 1
**Empirical Novelty And Significance:** 2
**Recommendation:** 3

**Strength And Weaknesses:**

Strengths:
1. The paper is clear and easy to understand.
2. The proposed model is straightforward and simple.

Weaknesses:
1. My biggest concern of this paper is novelty. All of the proposed three modules are not novel.
(1) Multimodal data isomorphism mechanism. As the most significant contribution of this paper, it is obvious that the idea of encoding time series data as images via GAF is inspired by Wang & Oates (2015). It is ok if only the motivation and basic logic is similar. However, I find out that the encoding method, the usage of notations, the equations, and even the figures are all very similar with the original GAF. The only difference is the normalization function in Eq. 5. I would like the other reviewers and editors to notice this problem, and the authors to provide a reasonable explanation about this problem.
(2) Multimodal sparse sampling. The sampling method proposed in the paper is just a simple extension of TSN (Wang et al. (2016)), where the authors additionally sample a GAF image from each segment. This extension is quite smooth and straightforward.
(3) Inter-segment modality attention mechanisms. This attention-based fusion method is rather simple and has been widely used in previous works for activity recognition [a, b].
[a] Meng, Lili, et al. "Interpretable spatio-temporal attention for video action recognition." In Workshop on ICCV, 2019.
[b] S. Sharma, R. Kiros, and R. Salakhutdinov, “Action recognition using visual attention.” In Workshop on ICLR, 2015.
2. Deep analysis is missing in the paper. The authors should give a clear comparison with SOTA methods not only on experimental performance but also on theoretical technology.
3. The performance comparisons are not satisfactory. The methods being compared in Tables 2 and 3 are out-of-date. The authors should compare with the state-of-the-art methods.
4. The inference details, the introduction to the datasets, and the qualitative results are missing in the paper.


**Summary Of The Paper:**

This paper presents a deep architecture based on RGB and IMU wearable sensors for multi-modal human activity recognition. The authors propose to encode the IMU sensor series into GAF images and then feed the RGB and GAF images into 2D CNNs for classification. Experiments are conducted on three public datasets.

**Summary Of The Review:**

The paper is clear and easy to understand. However, there are many issues in the current manuscript, such as the novelty and experimental comparisons. Therefore, I would recommend rejecting this paper in the current form.

---

### Decision · Program_Chairs · 2023-01-20

**Decision:**

Reject

**Justification For Why Not Higher Score:**

The ideas are not new and no deep insight provided on why or when it works or does not work. Authors did not response to all review comments

**Justification For Why Not Lower Score:**

NA

**Metareview: Summary, Strengths And Weaknesses:**

This paper proposes a deep architecture based on RGB and IMU wearable sensors for multi-modal human activity recognition with experimental results on three public datasets shown as comparable or better than SOTA.
The key weakness is novelty as it is mainly combination of known works. In particular, it relies heavily on the work by Wong and Oates (2015) but it is not necessary to spend more than a page just to describe GAFs in this previous work. Deep insights into why the approach performs better than the previous works will make the paper more appealing for acceptance


**Summary Of Ac-Reviewer Meeting:**

Not needed as 3 out of 4 reviews point to rejection while the last review only suggests marginal acceptance